# Facile Synthesis of Hollow Fe_3_O_4_-rGO Nanocomposites for the Electrochemical Detection of Acetaminophen

**DOI:** 10.3390/nano13040707

**Published:** 2023-02-12

**Authors:** Lazhen Shen, Jinlong Dong, Bin Wen, Xiangping Wen, Jianhui Li

**Affiliations:** College of Chemistry and Materials, Taiyuan Normal University, Jinzhong 030619, China

**Keywords:** acetaminophen, electrochemical detection, hollow Fe_3_O_4_ nanoparticles, magnetic Fe_3_O_4_-rGO/GCE electrode

## Abstract

Acetaminophen (AC) is one of the most popular pharmacologically active substances used as an analgesic and antipyretic drug. Herein, a new type of hollow Fe_3_O_4_-rGO/GCE electrode was prepared for electrochemical detection of AC through a three-step approach involving a solvothermal method for the synthesis of hollow Fe_3_O_4_ and the chemical reduction of graphene oxide (GO) for reduced graphene oxide (rGO) and Fe_3_O_4_-rGO nanocomposites modified on the glassy carbon electrode (GCE) surface. The as-prepared Fe_3_O_4_-rGO nanocomposites were characterized using a transmission electron microscope (TEM), X-ray diffraction (XRD), and a magnetic measurement system (SQUID-VSM). The magnetic Fe_3_O_4_-rGO/GCE electrodes were employed for the electrochemical detection of AC using cyclic voltammetry (CV), electrochemical impedance spectroscopy (EIS), and square wave voltammetry (SWV) and exhibited an ultra-high selectivity and accuracy, a low detection limit of 0.11 µmol/L with a wider linear range from 5 × 10^−7^ to 10^−4^ mol/L, and high recovery between 100.52% and 101.43%. The obtained Fe_3_O_4_-rGO-modified GCE displays great practical significance for the detection of AC in drug analysis.

## 1. Introduction

Acetaminophen (AC), also known as paracetamol, is one of the most commonly used antipyretic and analgesic drugs in clinical practice. It is used for fever, headache, joint pain, cancer pain and postoperative pain relief [1]. AC has a good curative effect and quick effect and basically has no adverse reactions at conventional doses. However, when the drug is overdosed, symptoms such as nausea, vomiting, stomach pain and other symptoms appear quickly, and liver damage will occur. An excessive dose will lead to liver failure, renal tubular necrosis, and even death due to renal failure [2,3]. In order to avoid the harm caused by AC to the human body, the accurate and efficient detection and analysis of AC components and content are particularly important. The traditional AC detection methods include high-performance liquid chromatography (HPLC) [4,5,6], liquid chromatography/mass spectrometry (HPLC/MS) [7,8], spectrophotometry [9], etc. These methods have high accuracy, but their equipment is expensive, and their detection rate is slow. It is of great significance to establish a convenient, sensitive and accurate analytical method for acetaminophen in vitro and in vivo. A promising alternative method in this respect is the electrochemical detection method, which possesses the advantages of high sensitivity, convenience, efficiency, and accuracy and exhibits great advantages in the detection of AC [10,11,12,13,14,15].

In recent research, nanocomposites were used in electrochemical applications, including as sensors, owing to their high surface area, high conductivity, and good electrocatalytic properties [3,16,17,18,19,20,21]. More recently, magnetic Fe_3_O_4_ nanoparticles have shown excellent magnetic properties, electrical conductivity, chemical stability, biocompatibility, and low toxicity. Therefore, Fe_3_O_4_ nanoparticles have received increasing attention as potential electrode modification materials or have been widely applied in electrochemical sensors and biosensors [22,23,24,25]. In addition, magnetic Fe_3_O_4_ nanoparticles with hollow structures can provide a larger electrochemically active area [26,27]. To further increase their electrochemically active area, hollow Fe_3_O_4_ nanoparticles were combined with reduced graphene oxide (rGO). Reduced graphene oxide consists of a layer of sp^2^-bonded carbon atoms that act as electron donor groups, thus exhibiting excellent electrical conductivity, large electrochemically active areas, and good chemical stability, and can be used as a high-quality conductor and carrier [3,28,29,30,31,32].

In this paper, we have prepared a new type of magnetic hollow Fe_3_O_4_-rGO nanocomposite-modified glassy carbon electrode (GCE) electrode for the detection of AC. The strategy for the process of preparing magnetic Fe_3_O_4_-rGO nanocomposites is depicted in Figure 1. The electrochemical behaviors of AC were investigated by different electrochemical characterization methods. Furthermore, the detection conditions, such as the type and pH value of the buffer solution, the amount of Fe_3_O_4_-rGO, waiting time, etc., were optimized. The detection method has the advantages of simple operation, high sensitivity, and good selectivity and is used for the detection of AC content in compound tablets, demonstrating a low detection limit and prominent repeatability.

## 2. Materials and Methods

### 2.1. Materials

Acetaminophen, graphene oxide (GO) and polyethyleneimine (PEI) were purchased from Aladdin Reagents (Shanghai, China). Iron chloride hexahydrate (FeCl_3_·6H_2_O), chromium trichloride (CrCl_3_·6H_2_O), sodium citrate, sodium carbonate, urea, polyacrylamide (PAM), hydrazine hydrate, potassium chloride (KCl), potassium ferricyanide (K_3_[Fe(CN)_6_]), potassium hexacyanoferrate(II) (K_4_[Fe(CN)_6_]), alumina (Al_2_O_3_) powders, glucose, uric acid, rutin, catechol, hydroquinone, and absolute ethanol were obtained from Tianjin Chemical Factory (China). The 0.2 mol/L phosphate buffer solution (PBS, pH 5.0, containing 0.9% NaCl) was used as the supporting electrolyte in the detection of AC, which was composed of 0.2 mol/L Na_2_HPO_4_ and 0.2 mol/L NaH_2_PO_4_. The standard solution of 0.01 mol/L AC was prepared using deionized water as a solvent. All chemical reagents and solvents used in this research were of analytical grade. Deionized water was used throughout the experiments.

### 2.2. Preparation of Hollow Fe_3_O_4_ Nanoparticles

Hollow Fe_3_O_4_ nanoparticles were prepared by a solvothermal method based on previous reports [33]. In the typical synthesis, 2.35 g of sodium citrate, 1.08 g of FeCl_3_·6H_2_O and 0.72 g of urea were dissolved in 80 mL of deionized water with strong stirring. After mixing evenly, 0.6 g of PAM was slowly added to the above mixture and further stirred for 30 min. Afterwards, the mixed solution was transferred into a Teflon-sealed autoclave and heated to 200 °C for 12 h. After cooling to room temperature, the obtained Fe_3_O_4_ nanoparticles were washed alternately with deionized water and absolute ethanol until the supernatant liquid was neutral and finally dried at room temperature in air for several hours.

### 2.3. Preparation of rGO

The rGO can be prepared by the reduction of GO with a reducing agent. Generally, 16 mg PEI was added to an aqueous GO solution (0.5 mg/mL, 20 mL), followed by ultrasonic dispersion for 5 min. Then, 400 µL of hydrazine hydrate was dropped into the mixed solution at 90 °C with stirring for 60 min. Furthermore, the solution was centrifuged and washed alternately with deionized water and absolute ethanol 3 to 5 times. Finally, the obtained sample was freeze-dried for 3 days to obtain rGO.

### 2.4. Preparation of Magnetic Fe_3_O_4_-rGO/GCE

Briefly, the as-prepared 9.7 mg rGO and 9.7 mg hollow Fe_3_O_4_ were dissolved in 9.7 mL of deionized water and dispersed by ultrasound for 1 h to form a homogenous mixture of hollow Fe_3_O_4_-rGO modification solution. The surface of the GCE electrode was polished with 0.05 µm Al_2_O_3_ water paste and then cleaned and dried to obtain a mirror-like surface (bare GCE). All of the above hollow Fe_3_O_4_-rGO modification suspension was dropped on the surface of bare GCE. After being dried under the irradiation of an infrared lamp, the modified electrode Fe_3_O_4_-rGO/GCE was obtained for all electrochemical experiments.

### 2.5. Characterization

Transmission electron microscopy (TEM, Hitachi Limited H-7650, Japan) was used to investigate the morphology and size of as-prepared nanomaterials. The crystal structures of the samples were measured using X-ray diffraction spectroscopy (XRD, Rigaku DMAX 2000 diffractometer with Cu Kα radiation, accelerating voltage = 40 kV, Tokyo, Japan). The magnetic measurement of the samples was carried out in a SQUID-VSM magnetic measurement system (Quant-um Design Company, USA).

### 2.6. Electrochemical Measurements

Cyclic voltammetry (CV) and square wave voltammetry (SWV) were performed on a CHI660E electrochemical workstation (Shanghai Chenhua instrument Co., Ltd. China) controlled with a conventional three-electrode system to test the electrochemical performance. The three-electrode system consisted of an auxiliary Pt wire electrode, a reference saturated calomel electrode and a hollow Fe_3_O_4_-rGO nanomaterial-modified GCE as a working electrode. For the CV method, the AC stock solution (100 µL, 0.01 mol/L) was diluted with 9.9 mL of PBS buffer (pH 5.0) to obtain 1 × 10^−4^ mol/L of AC test solution. The CVs were recorded in the range of −0.8 V to1.6 V at a scan rate of 100 mV s^−1^ up to 50 cycles for the stabilization and activation of the modified electrode. After each recording, the Fe_3_O_4_-rGO/GCE electrode was cleaned, re-polished to a mirror surface and then modified again. For the SWV method, 1 × 10^−5^ mol/L AC solution was prepared with PBS buffer solution in the electrolytic cell. The peak currents of 0.45 V for AC were recorded in the range of −0.2 V to 1.0 V at a pulse amplitude of 45 mV, frequency of 15 Hz, and scanning interval of 4 mV by the SWV method.

## 3. Results and Discussion

TEM and XRD were employed to characterize the prepared nanomaterials. Figure 1 shows the TEM images of the synthesized hollow Fe_3_O_4_ nanoparticles, rGO nanosheet and Fe_3_O_4_-rGO composites. As expected, the Fe_3_O_4_ nanoparticles, as shown in Figure 1a,b, are spherical, hollow, well-monodispersed and nearly uniform in dimension, with a particle size of about 400 nm. Figure 1c shows that the rGO obtained here exhibits the typical nanosheet-like structure with folded regions and transparent layers, indicating that the obtained rGO has low agglomeration and a low number of layers [34,35]. This structure is favorable for the attachment of hollow Fe_3_O_4_ on the surface of the rGO. It is obvious from Figure 1d that the hollow Fe_3_O_4_ nanoparticles are dispersed and anchored on the surface as well as on the folds of the rGO nanosheet. Compared with the hollow Fe_3_O_4_ and rGO, the hollow Fe_3_O_4_-rGO nanocomposite has a larger specific surface area.

The XRD patterns of Fe_3_O_4_ and Fe_3_O_4_-rGO are presented in Figure 2a. It can be seen that both hollow Fe_3_O_4_ nanoparticles and Fe_3_O_4_-rGO nanocomposites have five characteristic diffraction peaks at 2θ of 30.1°, 35.4°, 42.9°, 57.5° and 62.7°, corresponding to the (220), (311), (400), (422), (511) and (440) diffraction planes of magnetite, respectively [36,37], which indicates that the magnetite crystalline phase remained in hollow Fe_3_O_4_ and Fe_3_O_4_-rGO nanocomposites. Moreover, the characteristic diffraction peaks are sharp, suggesting that the synthesized samples have a high degree of crystallinity. Among the features of the hollow Fe_3_O_4_-rGO nanocomposites, a new weak broad peak at 2θ ≈ 22°–26° can be observed that corresponds to the (002) characteristic peak due to the reduction of functional groups containing oxygen in rGO [38,39,40]. All of the characteristic peaks of Fe_3_O_4_ and rGO observed in Fe_3_O_4_-rGO confirm the coexistence of both hollow Fe_3_O_4_ and rGO in the hollow Fe_3_O_4_-rGO nanocomposites.

The magnetic properties of samples were performed using the SQUID-VSM magnetic measurement system, and the magnetic hysteresis loops of Fe_3_O_4_ and Fe_3_O_4_-rGO are depicted in Figure 2b. As shown in Figure 2b, almost zero coercivity and residual magnetism in samples proved the superparamagnetic properties of hollow Fe_3_O_4_ nanoparticles and Fe_3_O_4_-rGO [37,41]. The bare Fe_3_O_4_ nanoparticles exhibited a high saturation magnetization (Ms) of 66.7 emu/g. Due to the introduction of the nonmagnetic rGO component, the saturation magnetization of Fe_3_O_4_-rGO decreased to 20.5 emu/g, which further confirmed the presence of rGO in the Fe_3_O_4_-rGO composites. The excellent magnetic properties of Fe_3_O_4_-rGO nanocomposites make them exhibit good magnetic separation performance.

Electrochemical impedance spectroscopy (EIS) measurement was carried out in 0.1 mol/L KCl solution containing 5 mM K_3_[Fe(CN)_6_] and 5 mM K_4_[Fe(CN)_6_] with an amplitude of 10 mV and frequencies ranging from 0.1 Hz to 10 kHz. The EIS can investigate the electrode interface, which can be modeled by an equivalent circuit. This equivalent circuit consists of four parts: the ohmic resistance of the electrolyte (*R*_s_), the electron transfer resistance (*R*_et_), the double-layer capacitance (*C*_dl_), and the Warburg impedance (*Z*_w_) [42]. The Nyquist plots of EIS for bare GCE, hollow Fe_3_O_4_/GCE, rGO/GCE, and Fe_3_O_4_-rGO/GCE are depicted in Figure 3a. The Nyquist plots include a semicircle part at high frequencies reflecting the electron transfer limited process at the electrode surface and a linear part at low frequencies corresponding to the diffusion process. The semicircle diameter of EIS is equivalent to the *R*_et_, which reflects the speed of electron transfer between the electrode surface and the solution of potassium ferricyanide. The smaller the semicircle diameter, the higher the conductivity and the faster the electron transfer [43,44]. As can be seen, the *R*_et_ value is 920 Ω on the bare GCE with the largest semicircle diameter. For hollow Fe_3_O_4_/GCE and rGO/GCE, the high electrical conductivity significantly decreased their *R*_et_ values to 450 Ω and 200 Ω, respectively. As shown in Figure 3a, the obtained Fe_3_O_4_-rGO/GCE exhibits electrical resistance as low as 180 Ω, lower than the values of both the single component of Fe_3_O_4_ and rGO, which is beneficial to rapid electron transfer for highly-sensitive AC detection [45].

The electrocatalytic behavior of different electrodes, namely of bare GCE, hollow Fe_3_O_4_/GCE, rGO/GCE, and Fe_3_O_4_-rGO/GCE toward 1.0 × 10^−4^ M AC, was studied by CV. As shown in Figure 3b, an oxidation peak appeared at a potential of 0.489 V when electrochemical measurements were performed with bare GCE electrodes. It is notable that the hollow Fe_3_O_4_/GCE electrode exhibits a higher oxidation peak than that of bare GCE, and the potential value decreased to 0.468 V. When the electrode was modified by rGO, the current of the oxidation peak was not only significantly higher than that of the hollow Fe_3_O_4_/GCE, but also, the potential value corresponding to the highest peak decreased to 0.427 V. In addition, a prominent reduction peak also appeared at 0.361 V. Compared with the rGO/GCE electrode, the peak current of the oxidation peak from the as-prepared Fe_3_O_4_-rGO/GCE electrode was further effectively enhanced at 0.424 V, and the peak shape became narrower. It can be seen that all four electrodes have a pair of redox peaks; among them, the redox peak of Fe_3_O_4_-rGO/GCE is the most obvious. It is further found that the redox peak current of the Fe_3_O_4_-rGO/GCE electrode is 16.99 µA, which is higher than 6.786 µA of bare GCE, showing that the presence of Fe_3_O_4_-rGO can greatly improve response sensitivity [45]. This may be due to the fact that when the hollow Fe_3_O_4_ nanoparticles are loaded on rGO, the unique hollow structure not only enhances the original specific surface area of rGO, but the synergistic effect of the two also further improves the electrocatalytic ability of the modified material to AC, thus showing an enhanced electrochemical signal [46].

Figure 4 shows the CV curves of 1 × 10^−4^ mol/L AC in PBS (pH = 5.6), Britton-Robinson (B-R, pH = 5.0), citric acid (CA)-sodium citrate (pH = 5.0) and HAc-NaAc (pH = 5.5) buffer solutions, respectively, with the hollow Fe_3_O_4_/rGO nanocomposite as a modified electrode. As can be seen from the figure, the peak shape of the oxidation peak of AC in PBS buffer is relatively narrower, and the peak current value is relatively higher, so PBS was chosen/selected as the buffer.

In the optimal buffer solution of PBS, the effect of solution pH on the peak current value of AC oxidation was further tested. The CV curves of AC on Fe_3_O_4_-rGO/GCE electrodes in PBS buffer solution with different pHs of 4.5, 5.0, 5.5, 6.0, 6.5, 7.0, 7.5, and 8.0 are depicted in Figure 5a. It is obvious from Figure 5a that the oxidation peak increases significantly with the pH of PBS increasing from 4.5 to 5.0. The current peak reaches its maximum at pH 5. When the pH value increases from 5.0 to 8.0, the current peak decreases rapidly. Additionally, the redox peaks shift towards negative potentials with increasing pH, indicating a strong dependence of the redox reaction on buffer pH value. The formal potential (*E*) is the average value of the oxidation peak and reduction peak and is plotted in Figure 5b as a function of pH. As can be clearly seen, with the increase in buffer pH, the potential corresponding to the highest point of the peak begins to shift negatively, and there is a good linear relationship between the pH value of PBS and the peak potential. The linear equation is: *E* = −0.0602 pH + 0.7505 (*R*^2^ = 0.9995). Since both sodium and disodium phosphate can exchange protons during ionization and hydrolysis, the electrode reaction involves protons. Changes in pH directly affect the degree of ionization or hydrolysis of H_2_PO_4_^−^ and HPO_4_^2−^. In order to obtain high sensitivity and an electrical signal, PBS buffer with pH 5.0 was finally selected as an electrolyte [47,48].

The effect of the modifier dosage and waiting time on the electrochemical signal was investigated by changing the modification amount and waiting time of the hollow Fe_3_O_4_-rGO composites on the electrode surface. The relationships between waiting time and modifier dosage with the oxidation peak current value are depicted in Figure 6a and Figure 6b, respectively. The CV method was used in the experiment, and the PBS buffer with pH = 5.0 was used as the supporting electrolyte to explore the relationship between the scanning time and the oxidation peak current value. As shown in Figure 6a, the peak oxidation current value initially increases with waiting time and reaches its maximum value at a waiting time of 60 s. When the waiting time is more than 60 s, the current value decreases rapidly. Therefore, the optimal waiting time was set to 60 s.

From Figure 6b, it can be seen that when the amount of modifier was increased from 4 μL to 6 μL, the electrochemical signal was enhanced obviously. When the amount of modification solution was more than 6 μL, the electrochemical signal was gradually weakened. The reason may be that the increase in the modifier thickened the Fe_3_O_4_-rGO modifier on the electrode, which hindered the transmission rate of electrons and affected the detection signal. Therefore, the optimal amount of hollow Fe_3_O_4_-rGO modification solution was selected as 6 μL.

Figure 7a shows the variation of specific capacitance from CV curves for the Fe_3_O_4_-rGO/GCE electrodes to detect AC with different scanning rates. An increase in the oxidation peak current value and the distortion of the shape of the CV curves can be observed with the increase in scanning rate in the scanning range of 60–220 mV∙s^−1^; meanwhile, the redox peaks shift towards positive potentials. The inset view of the same figure shows the fitting curve between the peak currents of AC and the square root of the scanning rates. It can be seen that the AC current value has a good linear relationship with the square root of the scanning rates. The linear regression equation is: *I*(µA) = 86.461$\sqrt{v}\;-$ 4.8931 (*R*^2^ = 0.9990), which indicates that the reaction process of AC on the hollow Fe_3_O_4_-rGO-modified GCE electrode is diffusion-controlled.

CV was carried out with a potential scanning range of −0.4−1.6 V, and the CV curves are given in Figure 7b. In order to determine the scanning range in this experiment, a final potential of 1.6 V was first chosen, and then the scanning range was narrowed by increasing the initial potential. As can be seen from the figure, the initial potential corresponding to the maximum current value is −0.4 V with increasing the initial potential from −0.6 V to 0 V. Furthermore, the initial potential was set to −0.4 V, and the scanning range was also narrowed by reducing the final potential. The maximum current value can be observed at the final potential of 0.8 V. This clearly demonstrates that the current peak is the highest, and the oxidation peak and reduction peak are symmetrical and prominent in the scanning range of −0.4−0.8 V. Therefore, the scanning range was determined to be −0.4−0.8 V during the experiment.

The SWV method was used to optimize the test conditions, and then different concentrations of AC were detected under the optimal conditions. The AC standard solution (200 µL, 1 × 10^−3^ mol/L) was diluted with PBS buffer (pH 5.0) to obtain a linear range of AC concentration of 5 × 10^−7^−1 × 10^−4^ mol/L. Figure 8 represents the SWV curves of AC with different concentrations from 5 × 10^−7^ to 1 × 10^−4^ mol/L. The peak was found to increase with an increase in AC concentration. The inset shows a good linear relationship between peak current and AC concentration. The linear regression equation is *I*(µA) = 0.5396 *c*(mol/L) + 1.1054 × 10^−6^ (*R*^2^ = 0.9957), and the detection limit is 0.11 µmol/L (signal-to-noise ratio S/N = 3). The analytical performances of Fe_3_O_4_-rGO-modified GCE were compared with other modified electrodes in the literature, showing a low detection limit, and the results are shown in Appendix A.

In the actual drug testing, some inorganic anions, cations and organic molecules that may be contained in drugs will affect the detection results. Therefore, research on the influence of interfering substances has been carried out, as shown in Figure 9. Under the optimal experimental conditions, 100 times the concentration of Na^+^, CO_3_^2−^, Cl^−^, glucose, uric acid, rutin, catechol, and hydroquinone and 50 times the concentration of Cr^3+^ were added to a 1 × 10^−4^ mol/L AC sample, respectively using the SWV method. Based on the detection results of AC as the standard, it can be seen that the above-added interfering substances have less interference on the detection results, which further indicates that hollow Fe_3_O_4_-rGO/GCE can automatically select signals belonging to AC in the presence of interfering substances.

The repeatability and stability of Fe_3_O_4_-rGO/GCE electrodes for AC detection were investigated. Under the optimal experimental conditions, the glassy carbon electrode was modified with 6 µL of hollow Fe_3_O_4_-rGO nanocomposite for the detection of AC. The parallel detection of 1 × 10^−4^ mol/L AC was repeated 5 times with the same Fe_3_O_4_-rGO/GCE electrode after scanning for 50 cycles by the CV method. The relative standard deviation (RSD) obtained was 1.53%. Additionally, the same electrode was re-modified consecutively 5 times with Fe_3_O_4_-rGO by the CV method for the determination of 1 × 10^−4^ mol/L AC under the same test conditions, and the obtained RSD was 3.03%. The RSDs indicate that the hollow Fe_3_O_4_-rGO-modified electrodes possess excellent repeatability and stability for AC detection.

In order to evaluate the practical applicability of the hollow Fe_3_O_4_-rGO nanocomposite-modified electrode for AC content in tablets, the acetaminophen tablets (Kangbide Pharmaceutical, specification: 0.5 g, actual mass: 0.609 g) were ground well into a fine powder and dissolved in deionized water to become a 1 × 10^−3^ mol/L AC solution. Then, 200 µL of the above AC solution was diluted in 4.8 mL PBS solution (pH 5.0) and diluted to 1 × 10^−5^ mol/L for electrochemical analysis. The acetaminophen concentration was determined from a calibration curve by averaging three repeated measurements. The recovery rate of AC is between 100.52% and 101.43%, and the RSDs are 0.32−1.39%. The results presented in Table 1 indicate that the hollow Fe_3_O_4_-rGO nanocomposite is a promising electrochemical platform for the accurate and reproducible detection of acetaminophen in drugs.

## 4. Conclusions

In conclusion, hollow Fe_3_O_4_-rGO nanocomposites were successfully prepared and modified on the electrode surface to research the properties of the composites and explore the detection ability of AC in this work. When the concentration of AC was within the range of 5 × 10^−7^–1 × 10^−4^ mol/L, the current value showed a good linear relationship with AC concentration, a low detection limit of 0.11 µmol/L, and a satisfactory recovery rate in the range of 100.52% to 101.43%. The experimental results showed that the Fe_3_O_4_-rGO/GCE electrodes can detect AC stably with excellent selectivity, accuracy, repeatability, and favorable anti-interference ability. This demonstrates that the Fe_3_O_4_-rGO/GCE electrodes can be used as a novel, low-cost and convenient electrochemical channel for the detection of AC content in acetaminophen tablets.

## Data Availability

Not applicable.

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
