# Peer review of "Facile Synthesis of Hollow Fe3O4-rGO Nanocomposites for the Electrochemical Detection of Acetaminophen"

_nanomaterials, 2023, doi:10.3390/nano13040707_

Round 1
Reviewer 1 Report
This contribution presents a new modified electrode applied to the measurement of material that has been frequently studied previously by electroanalytical techniques techniques. It contains little novel science but is of some interest and could be published. Further data on a range of matrices would have been desirable. Some values in the Tables are given to too high a number of decimal places
Author Response
We deeply appreciate the time and effort you have spent in reviewing our manuscript "Facile synthesis of hollow Fe3O4-rGO nanocomposites for the electrochemical detection of acetaminophen". We revised the manuscript, following your comments exactly.
Point 1: Some values in the Tables are given to too high a number of decimal places.
Response 1: Thank you for your reminder. Now the high decimal place values in the table have been reduced in the manuscript.
Reviewer 2 Report
This manuscript reports on the synthesis and application of a Fe3O4/r-GO nanocomposite as an electrochemical sensing material for a common drug, acetaminophen. Developing fast, easy-to-use and reliable sensors for drugs is of major technological and societal interest and the approach proposed here appears very interesting. The research strategy is solid and the results are convincing. The text is well structured and clearly written. In my opinion, this manuscript thus definitely deserves to be published. Only a couple of minor points need further justification/clarification:
- At a number of places in the text, the magnetic properties of the materials are pointed out. Is the magnetic character of those materials of any interest for the detection of acetaminophen as proposed here ? This should be clarified.
- To the general reader, it is not clear that the TEM images of Fig. 1 demonstrate that the Fe3O4 particles are hollow. This point should be better explained/justified.
- On page 8, a detection limit is determined. It would be interesting to compare that value with the detection limit of other analytical techniques used for acetaminophen.
- Technical point: is it reasonable to give RSD values with two decimal digits from a series of only five measurements (table 1) ?
To conclude, I suggest acceptance of this manuscript once these minor points have been addressed.
Author Response
We deeply appreciate the time and effort you have spent in reviewing our manuscript. The comments and suggestions are really thoughtful and helpful. Thus, we have modified the manuscript accordingly, and detailed corrections are listed below point by point:

Reviewer 3 Report
The paper can be published only after very major revision reflecting comments inserted as yellow notes into attached pdf. English must be improved.

Author Response
Thank you for your useful comments and suggestions on our manuscript. All the mistakes mentioned in attached pdf have been carefully corrected in the form of yellow notes in the manuscript. We hope the reviewers and the editors will be satisfied with the revisions for the original manuscript.

Round 2
Reviewer 3 Report
The paper can be published after very minor corrections reflecting yellow notes inserted into attached pdf of submitted manuscript. I believe they can be done by editor.

Author Response
Dear Reviewer,
We deeply appreciate the time and effort you have spent in reviewing our manuscript.
We have check and confirm the reviewer’s comments and found some small problems in the format, which were inserted into attached pdf of submitted manuscript.
